# Improving Effect of the Policosanol from *Ericerus pela* Wax on Learning and Memory Impairment Caused by Scopolamine in Mice

**DOI:** 10.3390/foods11142095

**Published:** 2022-07-14

**Authors:** Long Sun, Xian Li, Chenjing Ma, Zhao He, Xin Zhang, Chengye Wang, Min Zhao, Jin Gan, Ying Feng

**Affiliations:** Key Laboratory of Breeding and Utilization of Resource Insects, National Forestry and Grassland Administration, Institute of Highland Forest Science, Chinese Academy of Forestry, Kunming 650224, China; sunlong@caf.ac.cn (L.S.); leexian99@163.com (X.L.); machenjing@caf.ac.cn (C.M.); hezhao@caf.ac.cn (Z.H.); zhangxin@caf.ac.cn (X.Z.); wangcy@caf.ac.cn (C.W.); mzhao@caf.ac.cn (M.Z.); ganjin@caf.ac.cn (J.G.)

**Keywords:** *Ericerus pela* wax, policosanol, Alzheimer’s disease, Morris Water Maze, cognitive impairment, cholinergic system, oxidative stress

## Abstract

Policosanol (PC) is a mixture of long-chain fatty alcohols that exhibits multiple biological activities, such as reducing blood lipid and cholesterol levels, lowering blood pressure, and extenuating liver inflammation. To assess PC’s impact on cognitive behavior and function, PC was prepared from *Ericerus pela* wax using a reduction method and analyzed using gas chromatography (GC). A total of 60 mice were randomly divided into six groups of 10 animals each: control (0.5% CMC-Na solution, i.g.), model (0.5% CMC-Na solution, i.g.), donepezil (3 mg/kg, i.g.), PC low- (2 g/kg, i.g.), medium (4 g/kg, i.g.), and high- (6 g/kg, i.g.) dose groups. All the groups were administered daily for 28 consecutive days. There were four parameters—escape latency, crossings of platform, swimming distance, and time spent in the target quadrant—that were recorded to evaluate the cognitive performance of mice in the Morris Water Maze (MWM). After MWM testing, the levels of acetylcholine (ACh), acetylcholinesterase (AChE), superoxide dismutase (SOD), malondialdehyde (MDA), and glutathione (GSH) that were present in brain tissue were determined using assay kits. The GC data showed that PC consisted of four major components: tetracosanol (14.40%), hexacosanol (48.97%), octacosanol (25.40%), and triacontanol (4.80%). In the MWM test, PC significantly decreased the escape latency (*p* < 0.05) and increased the crossings of the platform (*p* < 0.05) and swimming distance (*p* < 0.05) and time in the target quadrant (*p* < 0.05) in rodents compared to that in the model group. Moreover, PC increased the levels of ACh, SOD, and GSH; inhibited AChE; and reduced MDA in the brain tissue of the tested animals. This is the first report to evaluate the efficacy of PC for cognitive behavior and function in animals. Our findings demonstrate that PC from *E. pela* wax is likely to exert an enhancing effect on learning and memory by promoting the cholinergic system and attenuating oxidative stress, which will provide a new insight into the efficacy of PC and expand its application in the food, nutraceutical, and beverage industries.

## 1. Introduction

As the number of elderly people grows rapidly globally, age-related concerns, primarily neurodegenerative disorders and diseases, are increasing as well. Neurodegenerative disorders are defined as the progressive loss of specific neuronal cell populations and are associated with autoimmunity, cellular degeneration, and genetic mutation [1]. Alzheimer’s disease (AD) is a neurodegenerative disorder that is characterized by the loss of memory, cognitive impairment, decline in language skills, difficulty with reasoning and communication, and eventually, the failure to perform activities throughout one’s daily life. Furthermore, AD is irreversible and is the most common cause and well-known form of dementia among the elderly populations worldwide, accounting for approximately 60–80% of all cases thereof [2]. It is estimated that the prevalence of this disease in 2015 was 44 million people worldwide, and that this figure is expected to double by 2050 [3]. AD is the third leading cause of disability and death among the elderly after cardiovascular disease, cerebrovascular disease, and malignant tumors [4]. Therefore, it is time for the world to take effective measures to ameliorate this situation.

Policosanol (PC) is a mixture that is primarily composed of saturated long-chain aliphatic alcohols with carbon atoms ranging from 20 to 34 [5]. The composition of the mixture varies widely, depending on the origin of the material and the method of preparation; however, it contains four major components: tetracosanol (C_24_H_49_OH), hexacosanol (C_26_H_53_OH), octacosanol (C_28_H_57_OH), and triacontanol (C_30_H_61_OH). Small amounts or traces of heptacosanol, nonacosanol, and dotriacontanol can also occasionally be found. Despite being widely distributed in plant germs, pericarps, kernels, seed coats, and leaves, PC was initially derived from sugarcane (*Saccharum officinarum* L.) wax by Cuban researchers [6,7]. Currently, commercial PC can be obtained from various other natural sources, such as from wheat germ oil, rice bran wax, and beeswax. Modern clinical studies have shown that PC has various physiological and pharmacological functions in animals and humans, including reducing blood lipid and cholesterol levels, lowering one’s blood pressure, inhibiting atherosclerosis, extending Pi-induced calcification, and lowering liver inflammation [8,9,10,11,12,13,14]. Recent data have demonstrated that PC suppresses testosterone-induced alopecia in mice by regulating hormone levels and inhibiting hair follicles into the regression phase at an earlier rate [15]. Owing to these beneficial biological functions and their non-toxic qualities (LD50 ≥ 5000 mg/kg) [16], PC has been widely used in the fields of health food, medicine, and cosmetics in many countries, including Japan, USA, Canada, and the Caribbean. In 2017, the National Health Commission of the People’s Republic of China added PC (from rice bran wax) to the list of novel food ingredients, providing an important legal guarantee for the deeper application of PC in the food industry of China [17].

In addition to beeswax, *E. pela* wax is another natural source of insect wax for PC because of its massive production in southern China. This wax is secreted by male insects of the *E. pela* Chavannes and covers the branches of the host plants [18]. Wax has a long history of use as a traditional medicine—either by itself or in combination with other herbs—in Chinese doctors’ prescriptions for the treatment of hemostasis, coughing, pain relief, wound healing, alleviating weakness, and restoring muscles [19]. Owing to its clinically confirmed safety and high stability, it has been documented in all versions of the Chinese Pharmacopoeia. Chemical composition analysis showed that the main components of insect wax were esters, hexacosyl hexacosanoate (C_25_H_51_COOC_26_H_53_), hexacosyl tetracosanoate (C_25_H_51_COOC_24_H_49_), and hexacosyl octacosanol (C_25_H_51_COOC_28_H_57_), with percentages of 55.16%, 22.36% and 16.65%, respectively. These three compositions accounted for 94.2% of the total wax [19,20]. Recently, there has been a new approach (reduction method) for the preparation of PC from insect wax with higher purity and yield compared to that of conventional saponification [21].

Herein, we investigated the effect of PC that was prepared from *E. pela* wax on scopolamine-induced learning and memory impairment in mice to assess PC’s impact on cognitive behavior and function, and explore the potential to expand the role of PC in the food, nutraceutical products, and beverage industries as additives or supplements, and further accelerate the sustainable development of special insect wax.

## 2. Materials and Methods

### 2.1. Materials and Chemicals

*E. pela* wax—semifinished, first grade—was purchased from the Emeishan Institute of Insect Wax in Sichuan Province, southwestern China.

Sodium hydroxide, hydrochloric acid, chloroform, tetrahydrofuran, and lithium aluminum hydride (analytical grade) were supplied by Shanghai Maclean Biochemical Technology Co., Ltd. (Shanghai, China) A total of four PC standards, tetracosanol, hexacosanol, octacosanol, and triacontanol, were obtained from Sigma-Aldrich (Burlington, MA, USA) with a purity of >98%. Scopolamine hydrobromide was acquired from APExBIO (Houston, TX, USA) and dissolved in sterile normal saline at a final concen-tration of 4 mg/kg. Donepezil was purchased from the American MCE Company (Kowloon, Hongkong) and dissolved in 0.9% sterile normal saline at a final concentra-tion of 3 mg/kg. Sodium carboxymethyl cellulose (CMC-Na) was purchased from Bei-jing Soleibao Technology Co., Ltd. (Beijing, China). Assay kits for the determination of acetylcholine (Ach), acetylcholinesterase (AChE), superoxide dismutase (SOD), malondialdehyde (MDA), and glutathione (GSH) were purchased from Nanjing Jiancheng Institute of Biological Engineering (Nanjing, China). Normal saline (0.9%) was obtained from Hebei Tiancheng Pharmaceutical Co. Ltd. (Cangzhou, China).

### 2.2. Animal Screening and Administration

Kunming mice (male, 20 ± 2 g, 5–6 weeks) were provided by Hunan Slac Jingda Laboratory Animal Co., Ltd. (Changsha, China, license number: SYXK [Xiang] 2019-0017). All the animals were kept under standard laboratory conditions (ventilated room, 24 ± 1 °C, [50 ± 10%] humidity, 12 h light/dark cycle) to acclimate for one week and had free access to water and food. All the procedures were performed according to “the Principles of the Care and Use of Animals” (China) and were approved by the Laboratory Animal Ethics Committee of the Institute of Highland Forest Science of the Chinese Academy of Forestry. Prior to testing, the mice were screened by placing each mouse on the platform for 20 s to form a reference memory. Next, the mice were released into the water from a fixed starting position, facing the tank wall. Mice that were located on the platform within 60 s were immediately removed from the tank. Those failing to find the platform within the allotted time were either placed on the platform again or guided to the platform and stayed there for an extra 20 s. After 10 min, a second screening was performed. If the mouse could still not find the platform, it was disqualified. A total of 60 eligible mice were screened. These 60 mice were randomly divided into six groups of 10 animals each, namely the control (0.5% CMC-Na solution, i.g., 0.2 mL/10 g), the model (0.5% CMC-Na solution, i.g., 0.2 mL/10 g), donepezil (3 mg/kg), and PC low- (2 g/kg, i.g.), medium- (4 g/kg, i.g.), and high- (6 g/kg, i.g.) dose groups. The doses of the low, medium, and high groups were based on the pretests. All the groups were administered i.g. daily for 28 consecutive days. The experimental procedure is illustrated in Figure 1.

### 2.3. Preparation and Quantification of PC from E. pela Wax

PC was prepared from insect wax using a previously described reduction method. Briefly, powdered *E. pela* wax (100 g) and reducing agent lithium aluminum hydride (7 g) were added to a 500 mL flask containing 100 mL tetrahydrofuran and refluxed at 85 °C in a water bath. After 5 h, the reaction solution was evaporated under reduced pressure at 40 °C to recover the organic solvent via a rotary evaporator. Distilled water (7 mL), sodium hydroxide solution (0.5 mol/L, 14 mL), and distilled water (21 mL) were then added to the flask. Finally, a hydrochloric acid solution (1 mol/L) was used to neutralize the residual mixture. After filtration, the filter cake (PC, 90.7 g) was rinsed three times with distilled water and dried overnight at 55 °C.

A gas chromatograph that was equipped with a flame ionization detector (Agilent 7809 B, Santa Clara, CA, USA) was used to quantify PC. The column was HP-5 (30 m × 0.25 mm × 0.25 μm) and other parameter settings were designated as follows: injector 320 °C, detector 340 °C, column oven starting at 200 °C (1 min) and increasing to 320 °C (10 min) at a rate of 5 °C/min, holding for 10 min. carrying gas, nitrogen; fuel gas, hydrogen; flow rate: 1.0 mL/min, hydrogen flow rate, 30 mL/min; and air flow rate, 300 mL/min. The PC sample and four standards (tetracosanol, hexacosanol, octacosanol, and triacontanol) were dissolved in chloroform to a final concentration of 10–200 mg/L and injected onto the top of the column, without splitting, at a volume of 2 μL after filtration through a 0.45 μm PVDF film. The composition identification of PC was based on its retention time in comparison with the individual standards. The PC composition was calculated from the peak areas of the four standard curves.

### 2.4. Morris Water Maze (MWM)

MWM with a video-tracking system was used to evaluate the performance of the mice. MWM testing included spatial acquisition trials (days 24–28) and probe trials (day 29). Specifically, the testing was carried out in a 120 cm diameter circular tank, 40 cm in height, which was filled with water at a temperature of 20–22 °C. The water tank was separated into four quadrants (I, II, III, and IV). The escape platform was placed in the third quadrant and submerged 1 cm below the water surface. The platform was fixed throughout the acquisition trials. A total of 20 min prior to the acquisition trials, each group was administered scopolamine (3 mg/kg, 0.1 mL/10 g), whereas the control group was administered an equal volume of 0.9% normal saline. After screening the animal, an acquisition trial was initiated by placing each mouse on the platform for 20 s. Subsequently, the mouse was released into the water from a fixed starting position, facing the tank wall. The time that was taken for each mouse to find the platform for the first time (latency) was recorded. The mice that were located on the platform within 60 s were immediately removed from the tank. Those not finding the platform within the allotted time were either placed on the platform again or guided to it and forced to stay there for an additional 20 s. The escape latency was set to 60 s. The mice were released into the water from different quadrants each day. The mice were trained in two trials daily for successive days (days 24–28).

A probe test was conducted 24 h after the last training session (day 29). It was modeled in the same manner as in the spatial acquisition trial. The mice were released from the quadrant opposite the initial location of the platform. During the probe test, the platform was removed from the tank, and the mouse was allowed to swim for 60 s. Escape latency, crossings of the platform, swimming distance, and the time spent in the target quadrant were reported as indicators of performance assessment.

### 2.5. Determinations of AChE, Ach, SOD, GSH, and MDA in Brain Tissue

After completion of the probe trials, the animals were euthanized by exposure to CO_2_ and cervical dislocation. The cerebral cortex and hippocampus tissues were then quickly separated on ice, rinsed with precooled normal saline (so as to remove the blood), cleaned with a filter paper, and weighed. A 10% homogenate was prepared by adding nine volumes of normal saline to the brain tissue sample and centrifuged at 1000× *g* for 20 min at 4 °C. The supernatant was maintained for determination of AChE, ACh, SOD, GSH, and MDA content, as per the manufacturer’s instructions.

### 2.6. Statistics

Excel software was used for the analysis of variance in the activity test. Experimental data are presented as the mean ± standard deviation. Individual differences between the groups were compared using LSD. A *p* < 0.05 indicates that the differences were considered statistically significant.

## 3. Results

### 3.1. Chemical Compositions of the PC

The data of gas chromatography (GC—Figure 2) showed that there were four major compositions—tetracosanol, hexacosanol, octacosanol, and triacontanol—in the prepared PC from *E. pela* wax via the reduction method with lithium aluminum hydride reducing agent. The composition percentages were 14.40%, 48.97%, 25.40%, and 4.80%, which accounted for 93.57% of the PC.

### 3.2. Acquisition and Probe Trials

As shown in Figure 3, during the acquisition testing (day 24–28, a five days), the latencies gradually decreased for the control group animals in their ability to locate the hidden platform, which indicated that repeated training was effective. In contrast, the model mice that were treated with scopolamine showed significantly higher latencies (*p* < 0.05) than those in the control group. In addition, the latencies of the donepezil group were significantly decreased (*p* < 0.05) compared to those of the model group, indicating that the impaired model of learning and memory was successful. The animals reached an asymptote on the fourth day (day 27) of testing. Compared to the model group, the mice in each group spent less time finding the platform, but there were no significant differences among them. This means that the different doses of PC had no effect on the initial learning task.

In the probe trial (day 29)—as shown in Figure 4—the escape latency of the model group mice was significantly higher than that of the control group (*p* < 0.01). Compared with the model group, the escape latency of the high-dose PC group was significantly decreased (*p* < 0.05). Compared with the model group, the crossings of the platform in the medium-dose PC group were significantly increased (*p* < 0.05). In addition, the swimming distance and time in the target quadrant of this group increased significantly (*p* < 0.05). These results suggest that PC could improve the cognitive behavior and function of the tested mice in an MWM.

### 3.3. Levels of AChE, Ach, SOD, GSH, and MDA

Compared to the model group, the AChE activities of the medium- and high-dose PC groups were significantly reduced (*p* < 0.05) (Figure 5a). Additionally, the ACh levels in the low- and medium-dose PC groups were significantly increased (*p* < 0.05, Figure 5b). The results showed that PC significantly suppressed the activity of AChE, resulting in an increase in the level of ACh in the brains of mice. Compared to the model group, the SOD activities of all the PC-treated groups, and the low-, medium-, and high-doses significantly increased (*p* < 0.05, Figure 5c). Furthermore, the GSH content of the low- and medium-PC groups was significantly increased as well (*p* < 0.05) (Figure 5d). In addition, the MDA levels in the high-dose PC group were significantly increased (*p* < 0.05) (Figure 5e). These findings showed that PC had positive effects on the biomarkers that are related to cognitive function in the brain tissue of the tested mice.

## 4. Discussions

As mentioned above, PC was first isolated in 1990 s from plant wax. Since then, numerous studies have explored its biological functions. It has always been considered a promising candidate for anti-lipidemic compounds worldwide because of its most attractive function: the ability to increase high-density lipoprotein cholesterol levels and decrease low-density lipoprotein cholesterol levels in animals and humans [14]. However, it remains unknown whether PC has an impact on learning and memory. To the best of our knowledge, this is the first study to evaluate the efficacy of PC on cognitive behavior and function in vivo in animals. Our results indicated that PC not only improved the performance of rodents in the MWM test, but that it also boosted the levels of several biomarkers that are linked to cognitive function in the brain tissue of the tested animals. Therefore, the application of PC in the food, nutraceutical, and beverage industries will broaden.

Although AD is an extremely complex multifactorial disease and given that there is no cure for it at the present time, several hypotheses have been put forward with regard to its cause. The cholinergic hypothesis is one such plausible and acceptable theory [22,23]. This theory suggests that cholinergic neurons in the central nervous system are heavily affected either by a loss of neurotransmitters (e.g., monoamine, ACh, noradrenaline) or by a loss of enzymes for choline synthesis and degradation (AChE, choline acetyltransferase, and butyrylcholinesterase). Cholinergic deficit may result in memory dysfunction or age-related cognitive symptoms which manifest via, or as, dementia [24,25]. Over the past decades, accumulating evidence has shown a close relationship between the degeneration of cholinergic neurons and the pathogenesis of AD in patients [26]. To date, four drugs have been approved by the Food and Drug Administration to mitigate AD progression and the clinical symptoms thereof [27]. Donepezil, rivastigmine, and galantamine are cholinesterase inhibitors, which increase the amount of neurotransmitters by restraining the hydrolysis of cholinesterase in the brain. By blocking the binding of ACh to M receptors, scopolamine may inhibit the transmission of brain signals and interfere with the formation of short-term memory and has been widely used in the model of mechanisms for AD and drug development. In our study, the medium-dose group (4 g/kg) of PC increased the content of ACh and reduced the activity of AChE in the brain of memory-impaired mice than that in the scopolamine-impaired group, indicating that, under these experimental conditions, PC could inhibit the synthesis of AChE directly, thereby reducing the decomposition of ACh. Hence, PC may be a potential AChE inhibitor for the adjuvant therapy of AD or neurodegenerative disorders.

Additionally, oxidative stress also plays an essential role in many neurodegenerative disorders such as AD and Parkinson’s disease (PD) [28]. Oxidative stress is defined as an imbalance between reactive free radicals and the antioxidant defense system. It can be induced by either an increase in free radical production, a decrease in antioxidant defenses, or both [29]. This antioxidant defense system contains antioxidants and enzymes that are involved in protecting cell components from oxidative stress damage [30]. As we know, free radicals are constantly produced via all types of physiological activities in our bodies. When excess free radicals accumulate, oxidative damage to the cells occurs. Their elimination largely depends on the antioxidant defenses. In particular, free radicals in the brain must be cleared efficiently and in a timely manner because the brain is extremely susceptible to oxidative damage due to its high levels of unsaturated lipids, oxygen, redox metal ions, and poor antioxidant systems [31]. In general, the brain weight just accounts for 2% to 3% of an adult’s body weight, but its oxygen consumption makes up more than 20% of the total oxygen consumption of the human body. Due to the high utilization of inspired oxygen in brain, many kinds of free radicals will inevitably be generated by a series of complex biochemical reactions. These free radicals include superoxide radical anion (O_2_·), hydroxyl radical (OH·), nitric oxide (NO·), peroxyl radical (ROO·), and so on [32]. Recently, several studies have illustrated that oxidative stress that is caused by free radicals is an early event in patients with AD. SOD is an endogenous antioxidant enzyme that converts O_2_·into hydrogen peroxide and O_2_, and its reduced activity can lead to oxidative damage in the brain. MDA is an important biomarker of lipid peroxidation and its abnormal increase is associated with memory impairment. GSH is an antioxidant molecule that helps to fight free radicals. The oral administration of octacosanol has been reported to alleviate disrupted hepatic reactive oxygen species metabolism that is associated with acute liver injury progression in rats [33]. More recently, PC was found to decrease the Aβ-induced paralysis rate and prolong the lifespan of *Caenorhabditis elegans* by modulating gene expression, including heat shock response, anti-oxidative stress, and glutamine cysteine synthetase [34]. Our findings provide further support for the protective role of PC against oxidative stress. Our data also suggest that PC improved SOD and GSH levels and decreased the level of MDA in the brain. Therefore, it is positive for the population over 60 years of age to monitor the markers of oxidative stress and, if necessary, supply them with PC-related antioxidant supplementation. Taking the aforementioned MWM results together, the PC from *E. pela* wax may probably exert the enhancing effect on learning and memory by boosting the cholinergic system (ACh, AChE) and attenuating the oxidative stress (SOD, GSH, MDA). These data imply that PC may have multiple targets or pathways against scopolamine-induced cognitive deficits, providing new insights into PC.

As analyzed in this study, the prepared PC comprises of tetracosanol, hexacosanol, octacosanol, and triacontanol. Compared to the plant origin, the PC from *E. pela* wax has a much greater percentage of hexacosanol (approximately 50%) and a lower percentage of triacontanol (~5%), whereas the former has a higher percentage of octacosanol (up to 70%). Previous reports have demonstrated that hexacosanol has neurotrophic properties in vivo, and intracerebrally-administered hexacosanol was able to promote the survival of forebrain cholinergic neurons after axonal injury [35]. In addition, hexacosanol promotes nerve regeneration [36]. Thus, a key question arises: which composition exactly contributes to the cognition-enhancing effect in rodents, one component alone, or four constituents synergistically? The findings of our study extended the application of PC in the neuroprotective field and suggested that the role of PC is no longer limited to neurotrophic effects. Commercial octacosanol can be used as a supplement for athletes because it increases running endurance time and improves biochemical parameters [37]. With the official approval by Chinese authorities as a novel food ingredient, PC can be used more widely as an additive and supplement in the functional food, nutraceutical, and beverage industries. Further research is required to determine whether the differences in compositions between PC from *E. pela* wax and the other existing PCs have a great influence on its biological properties.

Notably, there is barely any literature about the action mechanism of PC for its bioactive properties except for lowering lipids. Tetracosanol, hexacosanol, octacosanol, and triacontanol are homologues, which means they have similar structures and properties. In theory, most of linear alcohols, including PC, can be converted to aldehydes and acids by oxidation reactions. Kabir and his colleague reported the biodistribution of octacosanol in some organs and tissues after ^14^C-octacosano1 was administered, such as the liver, kidneys, spleen, heart, and lungs. They speculated that octacosanol may be partly oxidized and degraded to fatty acids through β-oxidation in rats [38], which gives support for the above oxidation theory. However, we still know little about what happens to PC after it enters the animal’s body and how PC works. Therefore, more work is needed to elucidate these issues.

## 5. Conclusions

To investigate whether PC can improve cognitive function, we prepared PC from *E. pela* wax and applied it to a mouse model of scopolamine-induced learning and memory impairment. Our findings indicate that the PC decreased the escape latency significantly, but markedly increased crossings of platforms, swimming distance, and the time spent in the target quadrant in the MWM test by rodents. Furthermore, PC increased the levels of ACh, SOD, and GSH, and reduced MDA and AchE activity in the brain tissue of the tested animals, suggesting that PC probably exerts a beneficial effect on learning and memory impairment by promoting the cholinergic system and attenuating oxidative stress. Our results will enrich our knowledge of PC and expand its application in the food, nutraceutical, and beverage industries.

## Figures and Tables

**Figure 1 foods-11-02095-f001:**
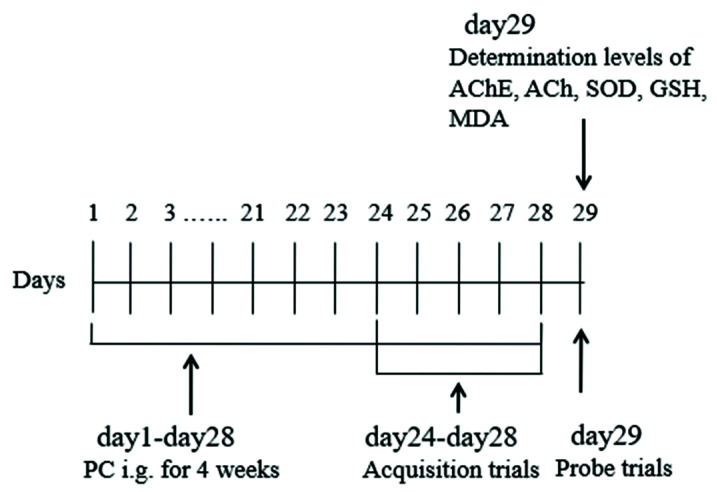
Schematic of the MWM testing and PC administration.

**Figure 2 foods-11-02095-f002:**
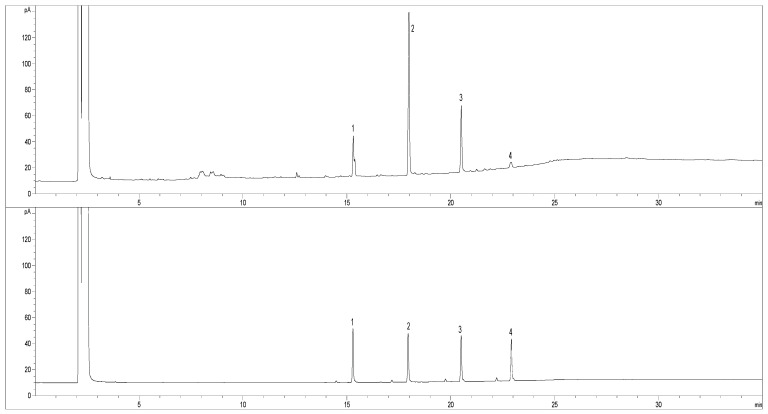
Gas chromatograms of PC that was obtained from *E. pela* wax (**upper**) and standard compounds (**bottom**). 1. tetracosanol; 2. hexacosanol; 3. octacosanol; 4. tricaontanol. The column was HP-5 (30 m × 0.25 mm × 0.25 μm) and the other parameter settings were designated as follows: injector 320 °C, detector 340 °C, column oven starting at 200 °C (1 min) and increasing to 320 °C (10 min) at a rate of 5 °C/min, holding for 10 min. carrying gas, nitrogen; fuel gas, hydrogen; flow rate: 1.0 mL/min, hydrogen flow rate, 30 mL/min; air flow rate, 300 mL/min; and sample volume: 2 μL without splitting. The PC sample was dissolved in chloroform and injected onto the top of the column after filtration through a 0.45 μm PVDF film.

**Figure 3 foods-11-02095-f003:**
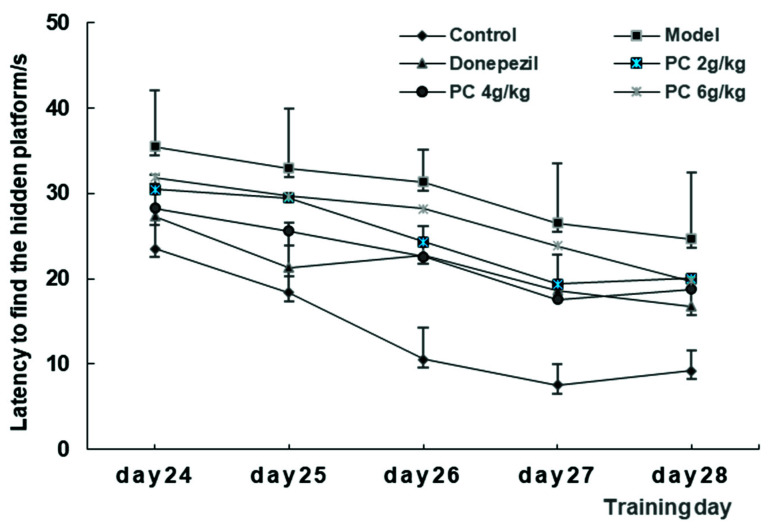
The latencies of mice in the acquisition trials of MWM. A total of 60 mice were randomly divided into six groups of 10 animals each, the control (0.5% CMC-Na solution, 0.2 mL/10 g), the model (0.5% CMC-Na solution, 0.2 mL/10 g), donepezil (3 mg/kg), and PC low- (2 g/kg), medium- (4 g/kg), and high- (6 g/kg) dose groups. All the groups were administered daily i.g. for 28 consecutive days. The acquisition trials lasted 5 days, day 24–day 28. The data represents the mean ± SD.

**Figure 4 foods-11-02095-f004:**
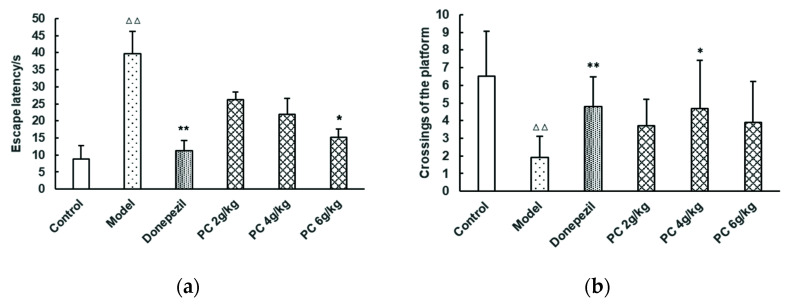
Mice’s performance in probe trials in the MWM (day 29). (**a**) escape latency, (**b**) crossings of the platform, (**c**) swimming distance in the target quadrant, and (**d**) the time spent in the target quadrant. A total of 60 qualified mice were randomly divided into six groups of 10 animals each, the control (0.5% CMC-Na solution, 0.2 mL/10 g), the model (0.5% CMC-Na solution, 0.2 mL/10 g), donepezil (3 mg/kg), and PC low- (2 g/kg), medium- (4 g/kg), and high- (6 g/kg) dose groups. All groups were administered daily i.g. for 28 consecutive days. The probe trials lasted 1 day after the acquisition (day29). The data represents the mean ± SD. (vs control: ^∆∆^, *p* < 0.01; vs. model: *, *p* < 0.05, **, *p* < 0.01).

**Figure 5 foods-11-02095-f005:**
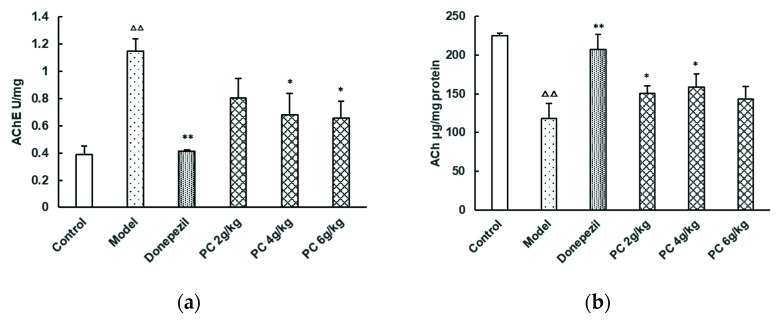
Levels of AChE, ACh, SOD, GSH, and MDA in the brain tissue of mice. (day 29). (**a**) AChE, (**b**) ACh, (**c**) SOD, (**d**) GSH, and (**e**) MDA. After the probe trails (day 29), the tested animals were sacrificed by exposure to CO_2_ and cervical dislocation, then quickly separated the cerebral cortex and hippocampus tissue on ice, rinsed with precooled normal saline, cleaned with a filter paper and weighted. A 10% homogenate was prepared by adding 9 volumes of normal saline into the brain tissue sample and centrifuged at 1000× *g* for 20 min at 4 °C. The supernatant was maintained for determination of the levels of AChE, ACh, SOD, GSH, and MDA using assay kits. The data represents the mean ± SD. (vs. control: ^∆∆^, *p* < 0.01; vs. model: *, *p* < 0.05, **, *p* < 0.01).

## Data Availability

The data presented in this study are available on request.

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
