# Peer review of "Improving Effect of the Policosanol from Ericerus pela Wax on Learning and Memory Impairment Caused by Scopolamine in Mice"

_foods, 2022, doi:10.3390/foods11142095_

Round 1

Reviewer 1 Report

This manuscript describes the memory enhancing property of policosanol obtained from an insect wax.

The study has been performed well with appropriate controls.

The authors need to consider the following:

Some minor English check will be required because some of the sentences are twisting the meaning of presentation. For example line 17 in abstract; Figure 1- determination levels of...

In line 37-39, authors are giving introduction about neurodegeneration. Based on the presentation, it looks like protein aggregation is the only reason for neurodegeneration. But it is not the only reason, there are other reasons too like autoimmunity, cellular degeneration...so these sentences need to be revised.

In figure 2, it would be better if the authors can include the peaks of the standard compounds described. Otherwise, from the current GC chromatogram, it is not evident whether their claim of compounds is valid.

There is also no clarity on how the dose for treatment was fixed. Authors need to include the logic for medium-high dose in this study.

In lines 284 - 287, authors say that the policosanol, induced production of ACh and reduced release of AChE for improving memory. Increased production and release of ACh can have other complications in muscle contraction and nervous transmission, and reduced AChE release can worsen this. Now, what justification authors have for claiming that policosanol does a balanced production and release of ACh and AChE to maintain both memory and optimal muscle contraction activities.

Reviewer 2 Report

The present study evaluated the impact of Policosanol (PC) from Ericerus pela wax on cognitive behaviour and function.

As referred by the authors, this is the first study assessing the efficacy of PC on cognitive behaviour and function in vivo in animals, thus its novelty and interest. At the same time, this study intends to value other sources of Policosanol, namely Ericerus pela wax, as the most common are wheat germ oil, rice bran wax, and beeswax. 

The article's writing is coherent and organized, being easy to understand. The only thing missing would be other studies' references to compare the results. However, since this appears as a new study, the lack of comparison is understood. I would also remove the description of the methodology from the figure captions since it is already in the respective section.

Reviewer 3 Report

This article is very interesting, especially because it relates food to the cognitive part. Some minor suggestions are made to the document.

It would have been interesting to know how much PC was bioavailable in the mice.  Why was PC not quantified in mice?

Lines 299-300. The brain is extremely susceptible to oxidative damage due to its high levels of unsaturated lipids, oxygen, redox metal ions. According to the above, what type of free radicals or ROS could then be generated in the brain? Discuss more about this.

Lines 321-338- It remains to discuss a little more the structure-activity relationship of the PC. How does the structure influence the determined biological activity?

What would be the mechanisms, types of bonds in the brain structures or biotransformation of PC to carry out its action? Discuss more about it.
